# GM-CSF Protects Macrophages from DNA Damage by Inducing Differentiation

**DOI:** 10.3390/cells11060935

**Published:** 2022-03-09

**Authors:** Tania Vico, Catrin Youssif, Fathema Zare, Mònica Comalada, Carlos Sebastian, Jorge Lloberas, Antonio Celada

**Affiliations:** 1Biology of Macrophages Group, Department of Cellular Biology, Physiology and Immunology, University of Barcelona, 08007 Barcelona, Spain; tania.7.91@hotmail.com (T.V.); catrin_y@yahoo.com (C.Y.); fatemeh.zare63@gmail.com (F.Z.); mcomalada@hotmail.com (M.C.); csebastian@ub.edu (C.S.); 2Institute of Biomedicine, University of Barcelona (IBUB), 08028 Barcelona, Spain

**Keywords:** macrophage, growth factors, proliferation, differentiation, DNA damage

## Abstract

At inflammatory loci, pro-inflammatory activation of macrophages produces large amounts of reactive oxygen species (ROS) that induce DNA breaks and apoptosis. Given that M-CSF and GM-CSF induce two different pathways in macrophages, one for proliferation and the other for survival, in this study we wanted to determine if these growth factors are able to protect against the DNA damage produced during macrophage activation. In macrophages treated with DNA-damaging agents we found that GM-CSF protects better against DNA damage than M-CSF. Treatment with GM-CSF resulted in faster recovery of DNA damage than treatment with M-CSF. The number of apoptotic cells induced after DNA damage was higher in the presence of M-CSF. Protection against DNA damage by GM-CSF is not related to its higher capacity to induce proliferation. GM-CSF induces differentiation markers such as CD11c and MHCII, as well as the pro-survival Bcl-2A1 protein, which make macrophages more resistant to DNA damage.

## 1. Introduction

Macrophages play a central role in immune response and tissue homeostasis. These cells have two functions in inflammation, initially being pro-inflammatory by destroying infectious agents and infected tissues and then being anti-inflammatory by repairing the damaged tissues [1]. In the early stages of inflammation, after interaction with inflammatory activators such as lipopolysaccharide (LPS) and interferon γ (IFN-γ), macrophages produce reactive oxygen species (ROS) through adaptation of mitochondrial respiration by mitofusin 2 [2]. ROS are a variety of molecules and free radicals derived from molecular oxygen that are essential mediators of the macrophage with a variety of functions [3]. ROS are a useful tool for eliminating infectious agents but they are also DNA-damaging agents, putting the survival of monocytes and macrophages at risk [4,5] and therefore blocking the tissue reconstruction phase, which can lead to chronic inflammation.

We showed that macrophages have systems to repair DNA breaks when they become activated or injured [5]. Following DNA damage a number of molecules are specifically induced in macrophages by pro- but not by anti-inflammatory activators, such as Trex1 [6], NBS1 [5] and SAMHD1 [7]. In fact, the malfunction of genes responsible for clearing free nucleic acid fragments inside cells leads to the accumulation of intracellular nucleic acids and activation of sensors of the innate immune system. These free nucleic acids in turn induce the production of type I interferons associated with pathological conditions included under the syndrome of interferonopathies [8] or the sterile chronic inflammation associated with aging that has been termed the “senescence-associated secretory phenotype”, or SASP [9].

ROS can cause several types of DNA damage ranging from base modifications to DNA backbone breaks in human biology and disease [10]. The cell responds to this damage by activating the DNA damage response pathway, leading to cell cycle arrest and thus providing enough time for efficient DNA repair [11]. Another important route of cell inactivation following the induction of DNA damage is apoptosis. A large number of specific DNA lesions that trigger apoptosis have been identified [12]. Moreover, the mechanism of action of many anticancer drugs involves promoting DNA damage-induced apoptosis [13]. Together, these mechanisms are crucial for preventing the replication and propagation of potentially deleterious mutations.

Highly proliferative cells are the most sensitive to DNA-damaging agents. Since tumor cells are rapidly dividing cells, many DNA-damaging agents have been used to treat cancer, including methylating agents, cross-linking chemotherapeutics, topoisomerase inhibitors and radiomimetic drugs [14]. In addition to tumor cells, normal cells can also be affected by DNA damage. Among these, those of the immune system are highly sensitive when exposed to DNA-damaging agents due to their high proliferative potential. 

Macrophages proliferate in the presence of growth factors such as M-CSF and GM-CSF, which are also involved in the differentiation and maturation of macrophages from bone marrow progenitors [15] and are important survival factors for monocytes and macrophages [16,17]. In fact, M-CSF and GM-CSF induce two different pathways in macrophages, one for proliferation, mediated by ERK, and the other for survival, which requires the PI-3K/Akt kinases and p21^Waf1^ [17]. M-CSF and GM-CSF are expressed under different situations. While the production of M-CSF is ubiquitous and constitutive in many different cells, GM-CSF is produced by activated leukocytes as part of the immune system’s response during inflammation [18].

Although previous studies have compared the different phenotypes induced in macrophages by M-CSF and GM-CSF [19,20], little is known about the effect of these growth factors on the DNA damage response in macrophages in particular. In the present study, we focused on how bone marrow-derived macrophages growing in the presence of M-CSF or GM-CSF respond to different DNA-damaging agents. We found that macrophages growing in the presence of GM-CSF were less susceptible to the effects of DNA-damaging agents compared to those growing in the presence of M-CSF. This is due to the induction of cellular differentiation produced by GM-CSF that is associated with the expression of pro-survival Bcl-2A1 protein, which makes macrophages more resistant to DNA damage. Our results unravel a key unexpected mechanism of GM-CSF that allows macrophages to survive DNA damage at inflammatory loci.

## 2. Materials and Methods

### 2.1. Reagents

Recombinant growth factors (M-CSF and GM-CSF) were purchased from Pepro Tech (Cranbury, NJ, USA). Unless specified otherwise, the concentrations of growth factors used were 10 ng/mL. The antibodies used were as follows: mouse anti-γ-H2AX (Ser 139), mouse anti-histone H1, rabbit anti-phospho-Akt (Ser 473), rabbit anti-α tubulin, mouse anti-p-p53 (Ser 15), rabbit anti-RPA, rabbit anti-RPA32/RPA2 (phospho S4 + S8), mouse anti β-actin, goat anti-mouse horseradish peroxidase (HRP)-conjugated and goat anti-rabbit HRP-conjugated (Appendix A). All other chemicals were of the highest purity grade available and were purchased from Sigma. Deionized water was further purified with a Millipore Milli-Q system Q-POD A10 (Merck KGaA, Darmstadt, Germany).

### 2.2. Mice 

BALB-c mice were purchased from Charles River Laboratories, Barcelona Spain; 6–8-week-old females were used. Animal use was approved by the Animal Research Committee of the University of Barcelona (number 2523).

### 2.3. Cell Culture 

Bone marrow-derived macrophages (BMDM) were isolated from 6-week-old mice as described previously [21]. The cells were cultured in plastic bacterial culture dishes (150 mm) in 40 mL Dulbecco’s modified Eagle’s medium (DMEM, Cultek, Madrid, Spain) containing 20% heat-inactivated Fetal Bovine Serum (FBS, GIBCO, Thermo Fisher Scientific, Waltham, MA USA) and 30% L-cell conditioned media obtained from supernatants of L-929 murine fibroblast cell line (ATCC identifier CCL-1) as a source of M-CSF and 100 U/mL penicillin, as well as 100 mg/mL streptomycin (Sigma-Aldrich, Merck KGaA, Darmstadt, Germany). The cells were incubated at 37 °C in a humidified 5% CO_2_ atmosphere. After 7 days of culture, a homogeneous population of adherent macrophages was obtained (>99% F4/80). The gating strategy for BMDM analysis is shown in Appendix A. To render cells quiescent, at 80% confluence, macrophages were deprived of M-CSF-conditioned medium for 16–18 h before treatment. 

### 2.4. RNA Extraction and Real-Time RT-PCR 

Total RNA was extracted, purified and treated with DNAse using the ReliaPrep RNA system kit (Promega, Madison, WI, USA), as recommended by the manufacturer. 400 ng of RNA was retrotranscribed to cDNA using the Moloney murine leukemia virus (MMLV) reverse transcriptase RNAse H Minus (Promega), following the manufacturer’s specifications. Quantitative PCR (qPCR) was performed using SYBR Green Master Mix (Applied Biosystems, Waltham, MA, USA), as recommended by the manufacturer. Non-retrotranscribed RNA samples were used as negative controls for each gene. When signal was detected in these negative controls (<32 Ct), the primer pairs used were discarded and replaced with alternative primers for the same gene. Furthermore, the amplification efficiency for each pair of primers was calculated by making a standard curve of serially diluted cDNA samples. Only the pairs of primers with an amplification efficiency of 100 ± 10% were used. Real-time monitoring of PCR amplification was performed in the ABI Prism 7900 Sequence Detection System (Applied Biosystems).

Data were analyzed by the ΔΔCt method [22] using Biogazelle Qbase+ software (Biogazelle, Gent, Belgium). Gene expression was normalized to three reference genes (i.e., housekeeping genes): *Hprt1*, *L14*, and *Sdha*. The stability of these reference genes was determined by checking that their geNorm M value was lower than 0.5 [23]. The primers used are described in Appendix A.

### 2.5. Proliferation Assay

Macrophage proliferation was measured by [^3^H]-thymidine incorporation as described previously [24]. Cells were deprived of M-CSF for 16–18 h and then 10^5^ cells were incubated in 24-well plates (Costar, Washington, DC, USA) for 24 h in DMEM and 20% FBS in the presence of the growth factor. After this period, the medium was replaced with medium containing [^3^H]-thymidine. After 6 additional hours of incubation, the medium was removed and the cells were fixed in ice-cold 70% methanol. After three washes, the cells were solubilized and their radioactivity was measured. Each experiment was performed in triplicate and the results were expressed as the mean ± SD. For cell counting we used a hemocytometer.

### 2.6. Cytometry 

The phenotypic analysis of macrophages was conducted by direct immunofluorescence using flow cytometry. Cell recovery from culture plates was facilitated by treatment with trypsin (Biological Industries, Cromwell, CO, USA). Cells were resuspended in PBS and incubated for 15 min with rat anti-CD16/32 (BD Pharmigen, San Diego, CA, USA) at 4 °C for 30 min to block nonspecific binding. Then, antibodies against different markers were used in conjunction with their respective isotype controls. The antibodies used for cytometry are shown in detail in Appendix A.

Samples were analyzed using a Gallios multi-color flow cytometrer instrument (Beckman Coulter, Brea, CA, USA) set up with the 3-lasers 10 colors standard configuration. Excitation of DAPI was conducted using a violet (405 nm) laser. Forward scatter (FS), side scatter (SS) and FL9 (450/40 nm) fluorescence emitted by DAPI were collected. Aggregates were excluded by gating single cells according to their area vs. peak fluorescence signal. DNA analysis (Ploidy analysis) on single fluorescence histograms was performed using Multicycle software (Phoenix Flow Systems, San Diego, CA, USA).

### 2.7. Apoptosis

Apoptotis was determined by incubating BMDMs with the Annexin V-FITC Apoptosis Detection Kit, following the manufacturer’s instructions. Live or viable cells (double negative), necrotic (DAPI positive), early (Annexin-V positive) and late apoptotic (double positive) cell populations were detected by flow cytometry (Appendix A). Dead cells included those in early and late apoptosis and necrotic cells. Viability was quantified by staining the cells with Crystal Violet staining solution (0.5%) (Sigma-Aldrich) [25].

### 2.8. Cell Cycle Analysis

The cell cycle was analyzed as described previously [26]. BMDMs (10^6^) were cultured in DMEM + 10%FBS in 12-well plates for 16 h. They were then left unstimulated or treated as specified for 24 h and then fixed with 95% ethanol. Next, cells were incubated with propidium iodide (PI) (Sigma-Aldrich) and RNase A (Sigma-Aldrich). Cell cycle distributions were analyzed on the basis of propidium iodide (IP) staining (G1, S and G2). Gating strategy for cell cycle analysis is shown in Appendix A.

### 2.9. Western Blot Protein Analysis 

To obtain total protein lysates, BMDMs (at least 10^6^ cells) were washed in cold PBS and lysed with TGH-NaCl (1% Triton X-100, 10% glycerol, 50 mM HEPES and 250 mM NaCl) plus protease inhibitors, as indicated previously [5,27]. Lysates were centrifuged to remove cellular debris. The protein concentration was determined using the Bradford Protein Assay (Bio-Rad laboratories, Hercules, CA, USA). Total protein lysates (50 mg) were separated by SDS-PAGE and transferred to polyvinylidene difluoride (PVDF) membranes using the iBlot2 system (Thermo Fisher, Waltham, MA, USA) and following the manufacturer’s instructions. Membranes were blocked for 1 h at room temperature in blocking buffer (5% dry milk in TBS-0.1% Tween 20) and then incubated with primary antibody in blocking buffer for 16 h at 4 °C. The concentration of antibodies is in Appendix A. The membranes were then washed three times × 5 min with TBS-Tween and incubated for 1 h at room temperature with the corresponding HRP-conjugated secondary antibody diluted 1:1000 in blocking buffer. After washing as before, ECL detection was performed, and the membranes were exposed to X-ray films. When necessary, band intensity was quantified using the open source image analysis software Fiji [28]. 

For histone H2AX western blot, an acid extraction of proteins was performed as described by the anti-phospho-H2AX antibody manufacturer. Briefly, cells were washed in cold PBS and lysed with lysis buffer (10 mM HEPES pH7.9, 1.5 mM MgCl_2_, 10 mM KCl, 0.5 mM DTT and 1.5 mM PMSF). Hydrochloric acid was added to a final concentration of 0.2 N and the cell lysate was incubated on ice for 30 min. After centrifugation at 11,000× *g* for 10 min at 4 °C, supernatants were dialyzed twice against 200 mL 0.1 M acetic acid for 1–2 h and three times against 200 mL H_2_O for 1 h, 3 h and overnight, respectively. 

For NBS1 western blot, a chromatin acid extraction was performed as described [5] with some modifications. After cell lysed, nuclei were collected by centrifugation at 2000× *g* for 5 min at 4 °C. The supernatant was discarded and the pellet resuspended in 2–5 volumes of ice-cold HCl 0.2N and incubated on ice for 20 min. Centrifuged at 2000× *g* for 10 min at 4 °C and the supernatant was neutralized with the same volume of Tris-HCl pH8 1 M.

### 2.10. DNA Damage Susceptibility Assay 

To analyze the susceptibility of macrophages to DNA damage we treated the cells with etoposide (Tocris, Ellisville, MO, USA) or hydrogen peroxide (Sigma-Aldrich). Unless specified otherwise, the concentrations of etoposide or hydrogen peroxide used were 50 µM and 250 µM respectively. Cells were washed and left in complete medium (DMEM + 10% FCS + M-CSF or GM-CSF at the indicated concentrations) for different periods of time as indicated. 

### 2.11. Test Fraction of Activity Released (FAR) Assay 

The number of double-strand breaks (DSBs) in cells treated with etoposide or hydrogen peroxide was monitored using the FAR assay [29]. After exposure to drug, the cells (10^5^) were kept on ice at all times, and all solutions added to the cells were ice-cold. The cells were centrifuged, resuspended in PBS supplemented with 0.2 mg/mL sheared herring sperm DNA and 56 mM β-mercaptoethanol to inactivate excess calicheamicin γ1 and incubated on ice for 5 min. The cells were then washed with PBS, and 150,000 cells were mixed with melted agarose (1.25% type VII in PBS with 5 mM EDTA) and transferred to a plug mold. The cells in the plug were then lysed at 4 °C for a minimum of 24 h in lysis buffer (25 mM EDTA, pH 8.5, 0.5% SDS, 3 mg/mL proteinase K added just prior to lysis). Longer incubation times did not alter the quality of the data. The cells were then resolved using agarose gel electrophoresis (0.7%) in 1× TAE (0.04 M Tris acetate, 1 mM EDTA, pH 8) at 4 °C for 17 h at 2 V/cm. The relative amount of cellular DNA migrating into the gel (FAR) was quantified using laser scanning equipment to calculate the number of DSBs.

### 2.12. Senescence-Associated β-Galactosidase Staining

The cells (10^5^) were washed in phosphate-buffered saline (pH 7.4) and fixed with 2% formaldehyde and 0.2% glutaraldehyde for 10 min at room temperature. After being washed twice, the cells were incubated at 37 °C for 4 h in a humidified chamber with freshly prepared staining solution (1 mg/mL X-Gal (Sigma-Aldrich) in dimethylformamide, 40 mM citric acid and phosphate buffer, pH 6.0, 5 mM potassium ferrocyanide, 5 mM potassium ferricyanide, 150 mM sodium chloride and 2 mM magnesium chloride). At the end of the incubation, the senescence-associated β-galactosidase staining rate was calculated by counting four random fields per dish and assessing the percentage of senescence-associated β-galactosidase staining-positive cells from 100 cells per field [30].

### 2.13. Quantification and Statistical Analysis

Data were analyzed using the unpaired Student’s t test, as indicated in each figure legend. When two or more variables were compared, a one-way ANOVA test followed by a Bonferroni correction was used, as indicated in the figure legends. Center, dispersion and n are defined in each figure legend. For all analyses, significance was set at *p* < 0.05. Statistical analyses were performed using GraphPad Prism 9.0 software.

## 3. Results

### 3.1. GM-CSF Induced Increased Protection against DNA Damage in Relation to M-CSF 

Bone marrow-derived macrophages are a homogenous population of non-transformed cells that require growth factors for proliferation and survival [17]. M-CSF is the most potent and specific growth factor for these cells [16]. However, GM-CSF also promotes macrophage proliferation and survival [17]. To determine the effect of these growth factors on the macrophage response to DNA-damaging agents, we first studied their susceptibility to treatment with etoposide, a topoisomerase II inhibitor used as an anticancer drug that causes DSBs [31]. Macrophages obtained from bone marrow cultures with M-CSF were starved of growth factor for 18 h and grown for 24 h in the presence of M-CSF or GM-CSF (Figure 1A). Then, macrophages were treated with the DNA-damaging agent etoposide for 1 h. After that, cells were washed and M-CSF or GM-CSF was added to the media, and 3 h later we analyzed the DSBs using the FAR test (Figure 1A). When we treated macrophages growing in the presence of M-CSF, etoposide induced DSBs in the DNA that increased in a dose-dependent manner when the amounts of the drug were increased (Figure 1A,B). However, when we cultured macrophages in medium containing GM-CSF, there was a significant reduction in susceptibility to this DNA-damaging agent (Figure 1A,B). 

We also used hydrogen peroxide to increase the oxidative stress in the cell, leading to base damage and DNA breaks [32]. This reagent also produced DSBs in macrophages, but in lower quantities than etoposide. Interestingly, again GM-CSF gave significantly better protection than M-CSF (Appendix A). 

To explore the effect of etoposide on single strand breaks (SSBs), we determined the expression of common markers such as phosphorylated replication protein A (RPA) [33]. While etoposide affected the macrophages incubated with M-CSF in a dose-dependent manner, no effect was observed in those incubated with GM-CSF (Figure 1C,D). Similar results were obtained when we used hydrogen peroxide instead of etoposide (Appendix A). Thus, the protective effect of GM-CSF seemed to be independent of the type of DNA damage. 

### 3.2. GM-CSF Induced More Rapid Recovery from DNA Damage Than M-CSF 

Next, we wanted to determine the influence of GM-CSF and M-CSF on the capacity to repair DSBs. The effect of etoposide on DNA breaks was assessed by detecting the induction of γ-H2AX, the H2A histone family member X (abbreviated as H2AX) that is a type of histone protein from the H2A family encoded by the H2AFX gene. An important phosphorylated form is γ-H2AX (S139), which forms when DSBs appear [34,35]. After treatment with etoposide for 1 h, cells were allowed to recover from DNA damage. After 3 h in the presence of GM-CSF, there was a significant decrease in γ-H2AX (Figure 2A,B). When we used hydrogen peroxide instead of etoposide, similar results were observed (Appendix A).

Macrophages were first treated with M-CSF for 24 h, then, after that, treated with etoposide for 1 h, washed and either incubated with M-CSF or GM-CSF (Figure 2C,D). Under these conditions, the damage caused by etoposide is the same for all cells that will later be incubated with M-CSF or GM-CSF. When we analyzed the recovery from DNA damage after 24 h there was also a significant difference between GM-CSF and M-CSF (Figure 2C,D). However, after 48 h in the presence of either GM-CSF or M-CSF, macrophages showed no increase in γ-H2AX, suggesting that in both cases the DNA was repaired. This experiment demonstrates that the ability to repair damaged DNA in cells incubated with GM-CSF is greater than in those incubated with M-CSF.

After 24 h of treatment with M-CSF or GM-CSF, the percentage of CD11b and F4/80 cells, as well as MHCII and CD11c was similar before and after treatment with etoposide or hydrogen peroxide. This data was confirmed when we measured the induction of phosphorylated p53, another marker associated with DSBs, after etoposide treatment [36]. There was a significant decrease in phosphorylated p53 in macrophages incubated with GM-CSF 3, 6 or 24 h after etoposide treatment, in relation to those incubated with M-CSF (Figure 2E,F). These results were confirmed when we used hydrogen peroxide instead of etoposide (Appendix A).

To confirm our results, we determined the expression of nuclear Nijmegen breakage syndrome (NBS1) protein. Together with the meiotic recombination 11 homolog (MRE11) and RAD50, NBS1 forms the complex MRE11, a DSB sensor that regulates the DNA damage response (DDR) and repair of DSBs [37]. Break detection by the MRE11 complex activates the ataxia telangiectasia mutated (ATM) kinase that promotes a robust DDR that includes the activation of checkpoint kinase 2 (CHK2) and the tumor suppressor p53 [38]. In macrophages treated with etoposide or hydrogen peroxide, NBS1 was obtained from the nucleus using chromatin acid extraction. The levels of phosphorylated NBS1 in cells incubated with GM-CSF after treating the cells with etoposide or H2O2 were lower in relation to the cells incubated with M-CSF (Figure 2G).

### 3.3. DNA Damage Induced Cell Cycle Impairment 

Exposure of eukaryotic cells to different DNA-damaging agents, such as ionizing radiation, UV light or reactive oxygen species (ROS), causes several types of DNA damage ranging from base modifications to DNA backbone breaks [10]. The cell responds to this damage by activating the DNA damage response pathway, leading to cell cycle arrest, which provides enough time for efficient DNA repair [11]. 

Both M-CSF and GM-CSF induce macrophage proliferation, but M-CSF is more powerful than GM-CSF. Thus, within 24 h under saturating concentrations M-CSF almost duplicates the number of cells, whereas GM-CSF induces lower amounts of proliferation (Figure 3A) [17]. However, when cells were treated with etoposide and subsequently cultured with growth factors there were fewer cells. Interestingly, under these conditions the number of cells treated with GM-CSF was higher than that of those treated with M-CSF (Figure 3A). Similar results were found when hydrogen peroxide was used instead of etoposide (Appendix A). These results are likely related to the capacity to repair DNA after genotoxin treatment. 

When there is abnormal processing of DNA, the replication checkpoints act as a brake and stop cycling, allowing DNA repair systems to correct replication errors [39]. In fact, if there is significant DNA damage that cannot be repaired successfully, cells will undergo apoptosis. However, if the DNA breaks are repaired, checkpoint signals will be attenuated and the cell cycle will be restarted and cells re-enter the cell cycle. The percentage of macrophages at the G2/M stage of the cell cycle increased after etoposide treatment, but there was a significant difference after this treatment when cells were incubated with M-CSF or with GM-CSF (Figure 3B). As shown herein, the damaging effect of etoposide prompts the expression of phosphorylated p53, a key regulator of the G1/S checkpoint [40], and induces the expression of stress sensors that modulate the response of mammalian cells to genotoxic stress such as growth arrest and DNA damage (e.g., Gadd45), which interact with other proteins implicated in stress responses [41]. The incubation of cells with etoposide induces the expression of the cyclin-dependent kinases (CDK) inhibitor *p21^waf−1^* and reduces that of *Gadd45* and *cyclin B1*, which explains the arrest of the cell cycle due to DNA damage (Figure 3C) [42].

### 3.4. GM-CSF Induced the Expression of Anti-Apoptotic B-Cell Lymphoma 2 (Bcl-2) A1 

Following the induction of DNA damage, an important route of cell inactivation is apoptosis. A large number of specific DNA lesions that trigger apoptosis have been identified [39]. These mechanisms are crucial for preventing the replication and propagation of potentially deleterious mutations. Because of DNA breaks after etoposide treatment, the number of apoptotic macrophages increased [43], and after 24 h of growth factor treatment, there was a significant difference between M-CSF and GM-CSF (Figure 4A). Similar results were obtained when hydrogen peroxide was used as a DNA-damaging agent (Appendix A). To determine the mechanism of apoptosis protection, we analyzed the levels of phosphorylated Akt, which is induced by M-CSF and GM-CSF for macrophage survival and is independent of the proliferation pathway [17]. With both treatments, phosphorylated Akt was induced after etoposide treatment (Figure 4B) or hydrogen peroxide (Appendix A). These results prompted us to measure the expression of genes whose products are involved in apoptosis after treating the cells with etoposide or hydrogen peroxyde. No differences were obtained between M-CSF or GM-CSF treatments when *Bcl-2*, *Bcl-X_L_* or *Bax* were measured; however, *Bcl-2A1* was notably induced by GM-CSF (Figure 4C). *Bcl-2A1* is a highly regulated NF-κB target gene that has important pro-survival functions [44]. In fact, *Bcl-2A1* was induced in macrophages that had been incubated with GM-CSF previously in response to etoposide damage (Figure 4D) and hydroxide treatment (Appendix A).

### 3.5. DNA Damage Was Independent of Proliferation 

Although M-CSF and GM-CSF are growth factors, macrophage proliferation was higher in the presence of M-CSF (Figure 3A). Because highly proliferative cells are more sensitive to DNA damage [45], we wanted to rule out the possibility that the protective effect of GM-CSF versus M-CSF was related to the decreased proliferative capacity elicited by macrophages in response to this growth factor. In a first attempt, we compared the macrophage proliferation dependent-induction of M-CSF and GM-CSF. We observed similar proliferation when we incubated the cells with 2 ng/mL of M-CSF or GM-CSF (Figure 5A). Then, we treated the cells with the same amount of growth factors and added etoposide. Under these conditions, the viability of macrophages was impaired in the presence of M-CSF in relation to GM-CSF independently of the amount of growth factor used (Figure 5B). The doses of growth factors did not affect the viability of cells treated with hydrogen peroxide (Figure 5B). Moreover, we treated macrophages growing in the presence of M-CSF with etoposide or hydrogen peroxide, and after washing we added either M-CSF or GM-CSF. Under these experimental conditions, we again observed a greater protective effect of GM-CSF over M-CSF (Figure 5C). Together, these results indicate that GM-CSF gives greater protection than M-CSF to bone marrow-derived macrophages against DNA-damaging agents, independently of the proliferative status of the cell. 

### 3.6. GM-CSF Induced CD11c^+^ MHC II^+^ Cells That Were Resistant to the Effect of Etoposide 

In the presence of GM-CSF, macrophages not only become activated but also express several molecules related to differentiation. GM-CSF activates the transcription factor PU.1, and a series of target genes, such as CD11b, mannose receptor, TLR4 and TLR2, are induced and their products expressed on the surface of macrophages [19,46]. After 72 h, more than 22% of macrophages incubated with GM-CSF, but not with M-CSF, expressed CD11c and MHC II, which are markers of dendritic cell differentiation (Figure 6A and Appendix A) [47]. GM-CSF and M-CSF differentially induce the expression of other markers. GM-CSF induces the expression of *Il-1β*, *Mannose receptor* and *Bcl2-A1*, while M-CSF induces *Tnf-α* and *Macrophage scavenger receptor 1* (Figure 6B). These opposing effects of M-CSF and GM-CSF on macrophages have been linked to “M2 and M1 macrophages” [48]. 

Finally, other macrophage surface markers, such as CD11b and F4/80, were expressed on almost 100% of macrophages, regardless of whether they were cultured with M-CSF or GM-CSF (Appendix A). The initial reduced percentage of F4/80 macrophages before the addition of M-CSF or GM-CSF is due to the 18 h deprived of M-CSF to render cells quiescent.

By staining with DAPI and annexin V we were able to differentiate necrotic, apoptotic and live cells (Appendix A). After 24 h of treatment with GM-CSF, the population that expressed CD11c and MHC II included significantly fewer necrotic and apoptotic cells in relation to the CD11c- and MHC II-negative population, before and after etoposide treatment (Figure 7A). Obviously, the more differentiated population contained more living cells. 

To confirm the GM-CSF-induced differentiation in macrophages we stained the senescence-associated beta-galactosidase, a marker of aging and differentiation (Figure 7B) [49]. There was a significant increase in cells treated with GM-CSF in relation to those treated with M-CSF, which express the lysosomal enzyme (Figure 7B). Finally, macrophages treated with GM-CSF expressed significantly higher levels of the cyclin-dependent kinases inhibitor p21^waf1^, which is associated with cellular aging [50] (Figure 7C).

## 4. Discussion

Bone marrow-derived macrophages are the equivalent of human macrophages in terms of origin, function, activation or proliferation [51,52,53], and they constitute the best available model for macrophage studies. 

The results presented show that GM-CSF is a better protector of macrophages against DNA damage than M-CSF. The capacity of GM-CSF to induce proliferation is less potent than that of M-CSF. However, the different strength of these two growth factors to protect against DNA damage is unrelated to their capacity to induce cell proliferation. 

In our experiments, we have treated macrophages with etoposide or with hydrogen peroxide. Etoposide, although it is not a physiological agent, produces more relevant DNA breaks in relation to hydrogen peroxide, making the results easier to quantify.

An interesting question is to determine whether these GM-CSF treated macrophages are actually resistant to DNA damage, or if they have increased/rapid repair capabilities. We have shown (Figure 2C) that the repairing mechanisms of cells submitted to the same injury previously treated with M-CSF are superior if the cells were treated later on with GM-CSF instead of M-CSF.

Another interesting question is the role of GM-CSF in DNA repairing. It has been shown in macrophages that GM-CSF induces ATM activating the Akt kinase [54], which is the key factor in the phosphoinositide 3-kinase (PI3K) pathway and regulates the activation of the major signals for cell growth, survival, metabolism and DNA repair [55]. Furthermore, monocytes accumulated DSBs following temozolomide treatment because they lack the DNA repair proteins XRCC1, ligase IIIa and PARP-1. Treatment with GM-CSF restored the expression of these proteins during differentiation into dendritic cells being able to repair DSBs [56].

Both M-CSF and GM-CSF activate two pathways upon interaction with their respective receptors in macrophages, inducing the same activities, one for proliferation, involving ERK, and the other one for protection against the induction of apoptosis associated with proliferation, which needs the PI-3K/Akt kinases and p21^Waf1^ [17]. However, M-CSF and GM-CSF induce two different differentiation programs in macrophages. Whereas M-CSF differentiates CD34 cells into macrophages [57], GM-CSF differentiates macrophages into “monocyte-derived” or “inflammatory” dendritic cells [58]. Therefore, upon engaging with their receptors, M-CSF and GM-CSF share the same pathways to induce proliferation and survival, but have different pathways to induce differentiation with diverse results. 

In this study, we used bone marrow-derived macrophages differentiated in vitro. These untransformed cells provide an excellent model that responds to growth factors as well as pro- (IFN-γ or LPS) and anti-inflammatory (IL-4) agents [59]. In our studies, two different DNA-damaging agents were used with similar results, although the effects of etoposide were stronger than those of hydrogen peroxide. Interestingly, not only does GM-CSF provide better protection against DNA damage, but it is also able to induce more rapid recovery of DSBs as monitored by the induction of γ-H2AX or phosphorylated p53. The reduction in proliferation that occurs following the DNA breaks was lower if the macrophages were treated with GM-CSF. This is due to the diminished induction of genes such as *Gadd45* or *Cyclin B1*, which code for proteins that block the cell cycle. Cellular proliferation is associated with DNA damage, being the basis for many chemotherapeutic agents used in the treatment of cancer or autoimmune diseases [60]. 

GM-CSF is an important factor that induces differentiation [61]. Our results showed that in macrophages incubated with GM-CSF for 24 h a population of cells emerged with the cell surface markers CD 11c and MHC class II, suggesting that they had become differentiated [19,46]. It has recently been shown that the exposure of bone marrow macrophages to GM-CSF induces the expression of CD11c and MHC class II molecules in two groups of cells that comprise conventional dendritic cells and monocyte-derived macrophages that behave as separable entities [47]. The data presented in that study do not invalidate our results since in both cases we demonstrated the differentiation produced by GM-CSF, which translated into an increase in CD11c and MHC II. We also showed that GM-CSF induces other differentiation markers, such as lysosomal senescence-associated β-galactosidase staining, a marker of aging and differentiation [49]. 

Interestingly, the more differentiated macrophages were the more resistant to induction of apoptosis/necrosis. After etoposide treatment and 24 h incubation with GM-CSF, fewer apoptotic cells were observed in relation to those incubated with M-CSF. Fascinatingly, when we tested different anti-apoptotic genes, only *Bcl-2A1* was induced by GM-CSF and not by M-CSF, confirming previous results [62]. Bcl2-A1 is a hematopoietic-specific protein and protects cells from apoptosis induced by a variety of apoptotic stimuli, such as DNA-damaging agents [63]. In response to GM-CSF, the *BCL2-A1* gene is induced and is a direct transcriptional target of nuclear factor NF–κB [44]. The transcription factor Spi-B regulates human plasmacytoid dendritic cell survival through direct induction of the antiapoptotic gene *BCL2-A1* [64]. 

BCL2A1 is able to reduce the release of pro-apoptotic cytochrome c from mitochondria and block caspase activation. The interaction of BCL2A1 with pro-apoptotic factors BAX and BAK has been described with conflicting results; however, it seems clear that BCL2A1 interacts with and blocks most of the BH3-only proteins that induce apoptosis through activation of BAX and BAK [65].

Our results could have implications for our understanding of the inflammatory process. Ly6C^high^ monocytes in an inflammatory locus differentiate into macrophages with a pro-inflammatory profile, releasing a number of molecules (IL-1α, IL-1β, TNF-α and IL-12) that activate helper T cells and induce the production of GM-CSF [66,67]. This cross-talk between macrophages and helper T cells, as well as the autocrine production of GM-CSF by monocytes [68], are crucial in inflammatory loci. On the one hand, GM-CSF induces macrophage differentiation of monocytes to inflammatory dendritic cells that play an important role in innate and adaptive immunity [69] and monocyte-derived macrophages that are double positive for CD11c and MHC class II. On the other hand, GM-CSF protects these two types of cell from DNA damage produced by the high levels of ROS. Although both macrophages and dendritic cells require mechanisms to repair DNA damage such as Trex1 [27,70], they also require the presence of GM-CSF to enhance their survival. Indeed, macrophages need to survive the anti-inflammatory phase to repair the tissue damage, and dendritic cells need to move from the inflammatory loci to the lymph nodes, carrying the antigens and initiating the acquired immune response. It has been reported that GM-CSF blockade in murine models of inflammation decreased the severity of the disease [71,72,73].

DNA breaks in the cells of the immune system and, particularly in macrophages, play a key role in immune-senescence [74]. It is interesting to note that macrophages from aged mice showed increased susceptibility to oxidants and an accumulation of intracellular reactive oxygen species [32], which could be responsible for DNA breaks [4]. Finally, telomeres shorten with age in macrophages, leading, as a result of decreased phosphorylation of STAT5a, to a reduced GM-CSF response [32]. Therefore, the increased susceptibility in aged macrophages to DNA breaks together with the reduced capacity of GM-CSF to repair the breaks may contribute to the reduced functional capacity of these cells in aging.

These observations, together with our results, emphasize the critical role of GM-CSF in protecting against DNA damage and ensuring the survival of macrophages in the homeostasis of the immune response.

## Figures and Tables

**Figure 1 cells-11-00935-f001:**
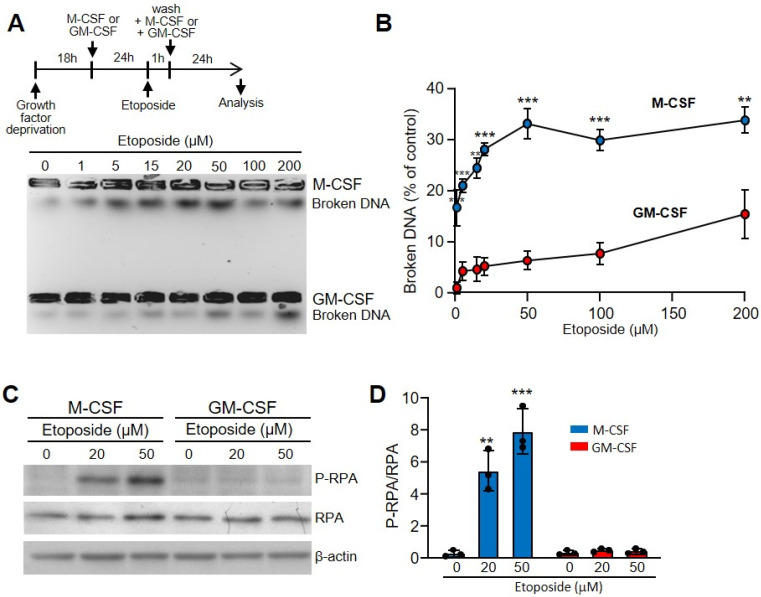
Etoposide induced more DSBs in macrophages incubated with M-CSF in relation to those incubated with GM-CSF. (**A**) The experimental design is shown at the top. Macrophages were obtained after 7 d of culture in the presence of M-CSF. Cells were deprived of M-CSF for 18 h and then incubated for 24 h with M-CSF or GM-CSF. Unless specified otherwise, the concentrations of growth factors used were 10 ng/mL. After that, etoposide was added for 1 h at the indicated concentrations, cells were washed and M-CSF or GM-CSF added again for 24 h before analysis. Representative results for DSBs determined using the FAR assay. (**B**) Quantification by densitometry of (**A**) (independent experiments, *n* = 3). (**C**) The experimental design is same as in A. Unless specified otherwise, the concentrations of etoposide and hydrogen peroxide used were 50 µM and 250 µM, respectively. SSBs were determined by immunoblotting of the phosphorylated RPA (RPA2, 32 kDa subunit), RPA (RPA1, 70 kDa subunit) and β-actin as controls. (**D**) P-RPA/RPA relation quantified by densitometry of C (*n* = 3). Each experiment was performed in triplicate, and the results are shown as the mean ± SD. ** *p* < 0.01 and *** *p* < 0.001 in relation to the corresponding treatments with M-CSF or GM-CSF, when all the independent experiments had been compared. Data were analyzed using Student’s *t*-test, with the exception of (**D**), which was performed using ANOVA test.

**Figure 2 cells-11-00935-f002:**
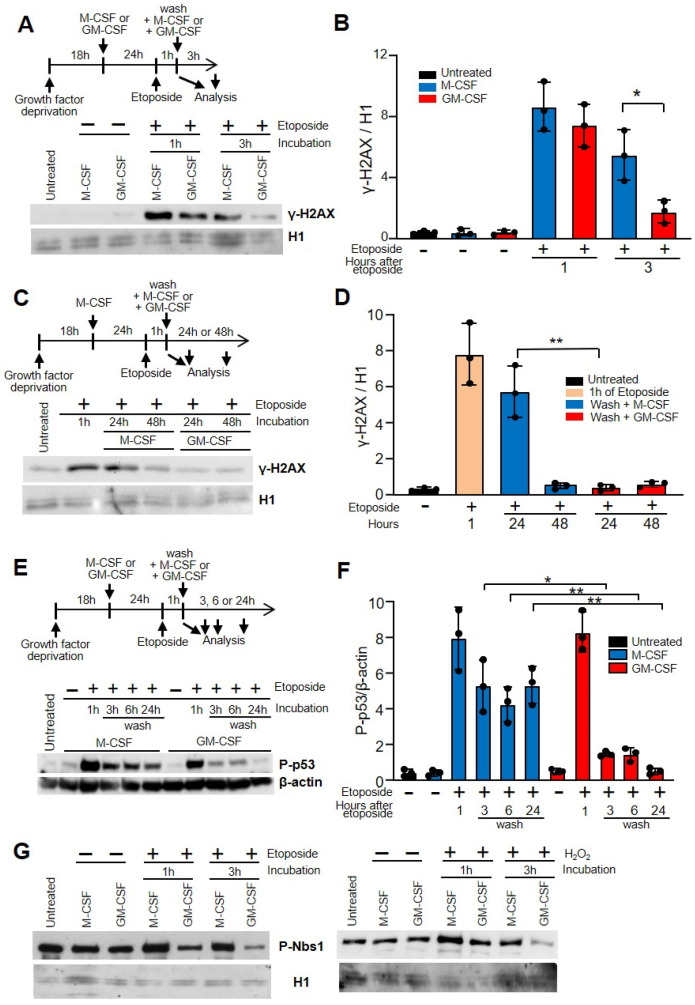
GM-CSF improved DNA damage repair in macrophages in relation to M-CSF. (**A**) The experimental design is shown at the top. After growth factor deprivation, macrophages were cultured with M-CSF or GM-CSF for 24 h. After that, etoposide was added for 1 h at 50 µM, cells were washed and M-CSF or GM-CSF was added for 3 h. DNA damage was determined by the induction of γ-H2AX immunoblotting using histone H1 as the control. ‘Untreated’ represents the cells 18 h after growth factor deprivation. (**B**) γ-H2AX/H1 relation quantified by densitometry of (**A**) (*n* = 3). (**C**) Similar experiment as in (**A**) but after etoposide treatment cells were washed and M-CSF or GM-CSF was added for 24 or 48 h. (**D**) γ-H2AX/H1 relation quantified by densitometry of (**C**) (*n* = 3). (**E**) Similar experiment as in (**A**) but samples were measured after etoposide treatment at 1, 3, 6 and 24 h. (**C**) is the control, comprised of cells treated with growth factors before etoposide addition. DNA damage was determined by phosphorylated p53 immunoblotting using β-actin as the control. (**F**) Quantification by densitometry of (**C**) (*n* = 3). (**G**) The experimental design is shown at the top. Macrophages were incubated for 24 h with M-CSF or GM_CSF, then treated for 1 h with etoposide or H_2_O_2_. Then, cells were washed and incubated again for 3 h with GM-CSF or M-CSF. Nuclear extracts were obtained by chromatin acid extraction. Each experiment was performed in triplicate, and the results are shown as the mean ± SD. * *p* < 0.05 and ** *p* < 0.01 in relation to the corresponding treatments with M-CSF or GM-CSF, when all the independent experiments had been compared. Data were analyzed using ANOVA test.

**Figure 3 cells-11-00935-f003:**
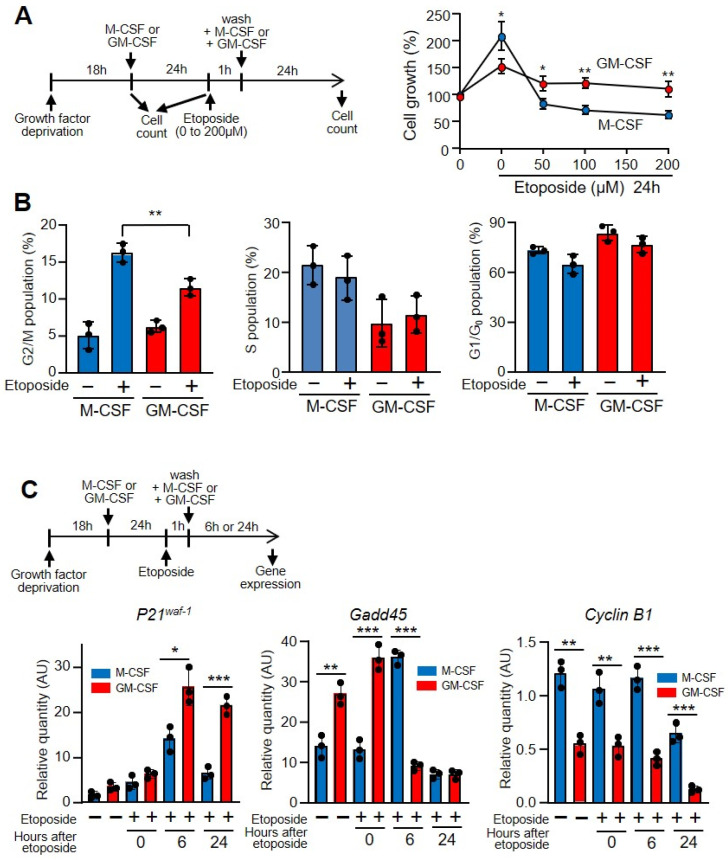
GM-CSF protected against the effects of etoposide on macrophage proliferation. (**A**) The experimental design is shown at the top. Macrophages were grown in M-CSF or GM-CSF (10 ng/mL) for 24 h, then treated for 1 h with the indicated amounts of etoposide, washed and incubated again with M-CSF or GM-CSF for 24 h, after which cell counting was performed using a hemocytometer (*n* = 3). (**B**) Macrophages were treated as in A in the presence of 50 µM of etoposide and their DNA content was measured by flow cytometry to determine the cell cycle (*n* = 3). (**C**) The experimental design is shown at the top. Macrophages were cultured in the presence of M-CSF or GM-CSF for 24 h, then incubated with etoposide for 1 h (50 µM), washed and incubated again for 6 or 24 h. Gene expression was analyzed by real-time PCR (*n* = 3). Each experiment was performed in triplicate, and the results are shown as the mean ± SD. * *p* < 0.05, ** *p* < 0.01 and *** *p* < 0.001 in relation to the corresponding treatments with M-CSF or GM-CSF, when all the independent experiments had been compared. Data were analyzed using Student’s *t*-test, with the exception of (**C**), which was performed using ANOVA test.

**Figure 4 cells-11-00935-f004:**
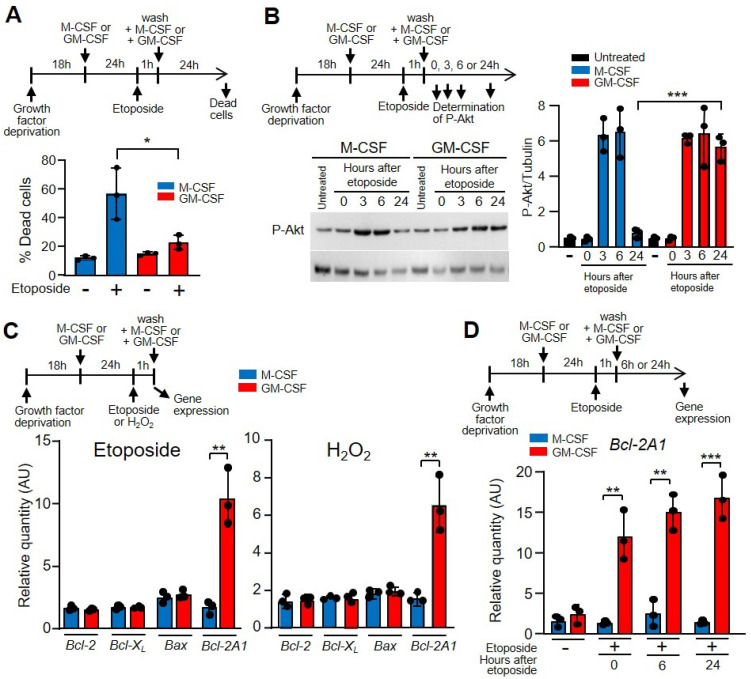
GM-CSF reduced the effect of etoposide on macrophage death. (**A**) The experimental design is shown at the top. Macrophages were incubated in M-CSF or GM-CSF for 24 h, then treated for 1 h with etoposide, washed and incubated again with M-CSF or GM-CSF for 24 h, after which dead cells (necrotic and apoptotic) were quantified by FACS analysis (*n* = 3). (**B**) The experimental design is shown at the top. The expression of phosphorylated Akt (^P^Akt) was measured by western immunoblotting using tubulin as the control (right panel). Left panel, quantification by densitometry of the western immunoblotting (*n* = 3). (**C**) The experimental design is shown at the top. Macrophages were cultured in the presence of M-CSF or GM-CSF for 24 h, then incubated with etoposide or H_2_O_2_ for 1 h, after which gene expression was analyzed by real-time PCR (*n* = 3). (**D**) The experimental design is shown at the top. Macrophages were incubated in M-CSF or GM-CSF for 24 h, then treated for 1 h with etoposide, washed and incubated again with M-CSF or GM-CSF for 6 or 24 h, after which gene expression was analyzed (*n* = 3). Each experiment was performed in triplicate, and the results are shown as the mean ± SD. * *p* < 0.05, ** *p* < 0.01 and *** *p* < 0.001 in relation to the corresponding treatments with M-CSF or GM-CSF, when all the independent experiments had been compared. Data were analyzed using Student’s *t*-test, with the exception of (**A**,**D**), which was performed using ANOVA test.

**Figure 5 cells-11-00935-f005:**
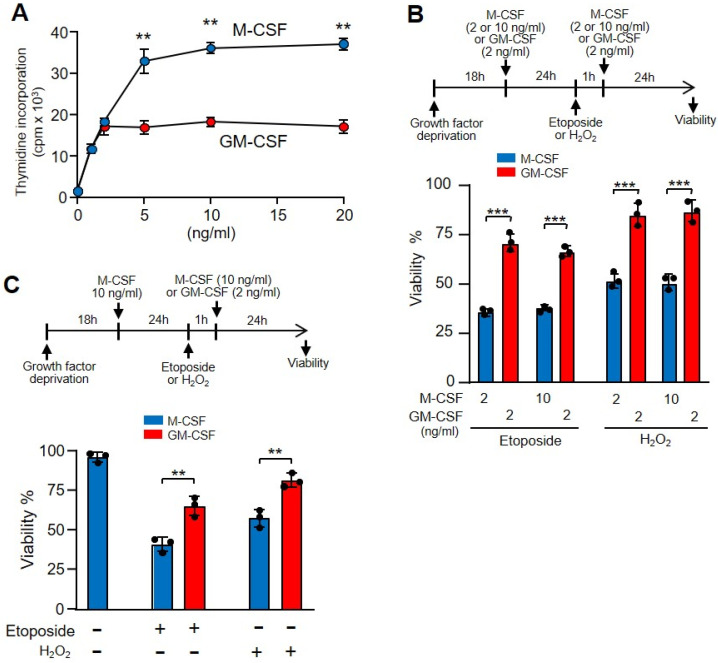
The different proliferation activity induced by GM-CSF or M-CSF was unrelated to the effect of etoposide. (**A**) Macrophages were incubated with M-CSF or GM-CSF at the indicated concentrations for 24 h and proliferation was measured as [^3^H]-thymidine incorporation (*n* = 3). (**B**) The experimental design is shown at the top. Macrophages were incubated in M-CSF or GM-CSF at the indicated concentrations for 24 h then treated for 1 h with etoposide or H_2_O_2_, washed and incubated again with M-CSF or GM-CSF for 24 h at the indicated concentrations, after which viability was quantified using Crystal Violet (*n* = 3). (**C**) The experimental design is shown at the top. Macrophages were incubated in M-CSF (10 ng/mL) for 24 h, then treated for 1 h with etoposide or H_2_O_2_, washed and incubated again with M-CSF (10 ng/mL) or GM-CSF (2 ng/mL) for 24 h, after which viability was quantified using Crystal Violet (*n* = 3). Each experiment was performed in triplicate, and the results are shown as the mean ± SD. ** *p* < 0.01 and *** *p* < 0.001 in relation to the corresponding treatments with M-CSF or GM-CSF, when all the independent experiments had been compared. Data were analyzed using Student’s *t*-test, with the exception of (**B**,**C**), which was performed using ANOVA test.

**Figure 6 cells-11-00935-f006:**
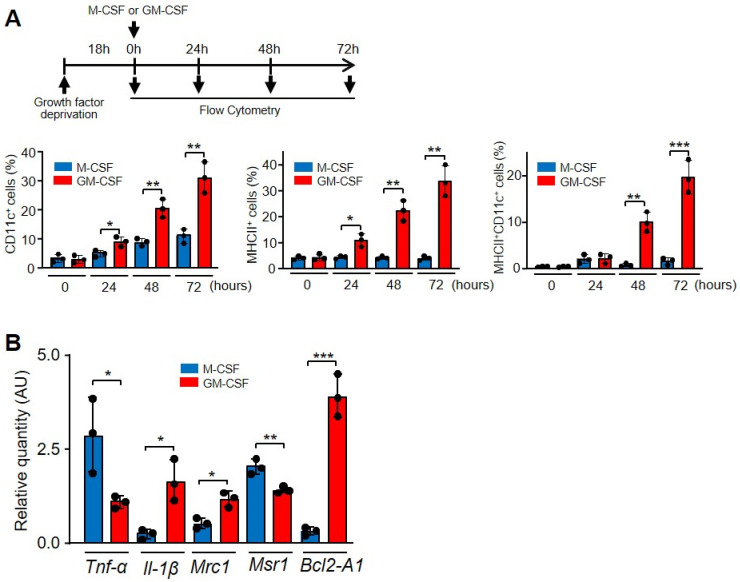
GM-CSF induced a different phenotype of macrophages in relation to M-CSF. (**A**) Characterization of CD11c and MHC II expression. The experimental design is shown at the top. After growth factor deprivation, macrophages were cultivated in the presence of the indicated growth factors for the indicated times and the surface expression was determined by FACS analysis and the corresponding antibodies. The graphics show the cells expressing single (CD11c or MHC II) or double markers (*n* = 3). (**B**) Macrophages were cultured in the presence of M-CSF or GM-CSF for 24 h and the indicated gene expression was analyzed by real-time PCR (*n* = 3). Each experiment was performed in triplicate, and the results are shown as the mean ± SD. * *p* < 0.05, ** *p* < 0.01 and *** *p* < 0.001 in relation to the corresponding treatments with M-CSF or GM-CSF, when all the independent experiments had been compared. Data were analyzed using Student’s *t*-test.

**Figure 7 cells-11-00935-f007:**
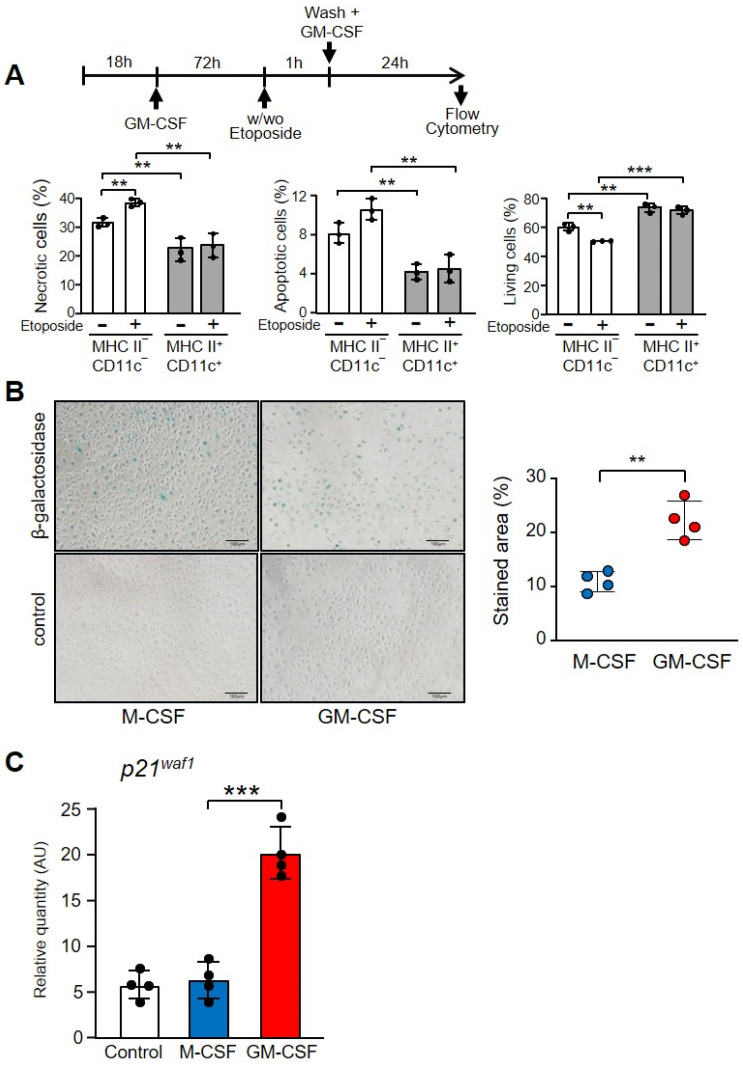
Differentiation markers of macrophages incubated with M-CSF or GM-CSF. (**A**) Necrotic, apoptotic or living cells in the population of macrophages treated with GM-CSF, comparing the phenotype MHC II^−^ and CD11c^−^ with MHC II^+^ and CD11c^+^. The experimental design is shown at the top. Macrophages were cultivated in the presence of GM-CSF (10 ng/mL) for 72 h, then treated or not with etoposide for 1 h and, after washing, GM-CSF (10 ng/mL) was added for 24 h. An example of quantification is shown in Appendix A. Gating was conducted using antibodies to CD11c and MHC II with the corresponding controls. Then, the double positive and negative cells were gated and DAPI and Annexin V staining was determined (*n* = 3). (**B**) Macrophages were incubated with M-CSF or GM-CSF for 24 h and then stained for senescence-associated β-galactosidase staining. On the left is an example of staining. The control was unstained cells. The right panel shows the quantification of four independent experiments. (**C**) Macrophages were cultured in the presence of M-CSF or GM-CSF for 72 h. Gene expression was analyzed by real-time PCR (*n* = 3). Each experiment was performed in triplicate, and the results are shown as the mean ± SD. ** *p* < 0.01 and *** *p* < 0.001 in relation to the corresponding phenotypes or treatments with M-CSF or GM-CSF, when all the independent experiments had been compared. Data were analyzed using Student’s *t*-test.

## Data Availability

All relevant data is available from the authors upon reasonable request.

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
