# Peer review of "GM-CSF Protects Macrophages from DNA Damage by Inducing Differentiation"

_cells, 2022, doi:10.3390/cells11060935_

Round 1
Reviewer 1 Report
Their results of this study show that GM-CSF is a better protector of macrophages against DNA damage than M-CSF and it allows macrophages to survive DNA damage at inflammatory loci, which could promote chronic inflammation. They also show that, while the capacity of GM-CSF to induce proliferation is less potent than that of M-CSF, the ability to protect against DNA damage is unrelated cell proliferation. Thus, they showed that the number of apoptotic cells induced after DNA damage was higher in the presence of M-CSF. Protection against DNA damage by GM-CSF was not related to its higher capacity to induce proliferation, but rather GM-CSF induces differentiation markers such as CD11c and MHCII, as well as the pro-survival Bcl-2A1 protein, which make macrophages more resistant to DNA damage. This is an interesting study and, for the most part, very well written. The Discussion clearly cites important papers and provides a nice summary of how this could promote inflammation in the context of aging/senescence or in inducing chronic inflammation under unchecked inflammatory conditions.
MAJOR COMMENTS:
- Methods and Results: All of the data was analyzed using the unpaired Student’s t test. However, in many of the figures there are more than two groups included and should be compared using ANOVA. For example, in Fig. 1D there are more than two groups that were treated with different doses. Another example is the evaluation of hours after treatment such as in Fig. 4B and Fig.4D. An ANOVA could tell you if there is a quantitative time-dependent effect and whether there is a significant increase vs. the zero time point. The same holds true for Fig. S1(E) and so on…
- Methods and Results: The following figures need to be assessed regarding statistical differences using ANOVA: Fig. 1D; Fig. 2B, Fig. 2D, Fig. 2F, Fig.3C (all three figures), Fig. 4A, Fig.4D, Fig. 5B, Fig. 5C, and the Supplemental Fig. S1 due to the reasons mentioned in the previous comment.
- Results: According to the Methods: Cell culture, Section 2.3., “After 7 days of culture, a homogeneous population of adherent macrophages was obtained (>99% F4/80).” The authors should show this in the supplemental figure section. Also, the authors should comment on why they see such a low percentage of F4/80+ cells in the Supplemental Fig. S3.
- Results: In four different places in the results, there is an incorrect reference to the supplemental figures, i.e., the text does not coincide with the results shown and the figure legend #. It should be revised accordingly in Lines 390-391, Lines 416-419, Lines 435-437, and Lines 438-439.
MINOR COMMENTS:
- Methods: The authors need to go through each section of the Methods to ensure that the a) concentrations, b) company info, and c) the composition of the media are included.
- Methods: Cell culture, Section 2.3.: The authors used 30% L-cell conditioned media as a source of M-CSF. Company info or fibroblast cell source and citations for this sentence are missing. The authors should also mention that the conditioned media is from the L929 murine fibroblast cell line.
- Methods: 5. Proliferation assay: The concentration of M-CSF and company info is missing in the following sentence (“Cells were deprived of M-CSF for 16–18 h and then 105 cells were incubated for 24 h in complete medium in the presence of the growth factor”) as well as the composition of the complete medium and where it was purchased. Additionally, which size/types of plates were used?
- Methods: Was the same concentration of M-CSF and GM-CSF consistently used in all experiments? The concentration and sources are missing in all of the sections and really should be included.
- Methods: The concentration of etoposide (Tocris, Ellisville, MO) or hydrogen peroxide should also be included.
- Discussion: The authors used mouse bone marrow-derived macrophages. Are there any expected differences or with human bone marrow-derived macrophages? Has anyone looked at these aspects in human blood- or bone marrow-derived monocytes? And are there any limitations in using mouse cells?
Author Response
Thank you, we have responded in the file, please see the file.

Reviewer 2 Report
The manuscript by Vico et al. aimed to elucidate how bone marrow-derived macrophages growing in the presence of M- CSF or GM-CSF respond to different DNA-damaging agents. This is a well-prepared and interesting study; however, some questions need to be discussed or experimentally addressed:
- First of all I would like to recommend the authors reorganize the paper: Fig.6 should be the figure. I would like the authors to also to show the data regarding the purity of obtained macrophages based on their morphology and surface markers.
- Fig.1 did the authors check the activation of ataxia–telangiectasia mutated (ATM) kinase that is activated when cells are exposed to DNA double-strand breaks. It will be good to provide such data.
- It will be good to confirm the data with Single Cell Gel Electrophoresis Assay in both neutral and alkaline conditions.
- Fig. 1- 2 what was the phenotype of macrophages after treatment with etoposide in both conditions with M-CSF and GM-CSF?
- Fig. 3 Please provide a gating strategy for FACS data.
- Fi.7 Did the authors look at p21 or p16 protein level? If not it will be a good confirmation of the pathway related to the induction of senescence.
Author Response

(The authors gave the same response as above.)

Reviewer 3 Report
In the manuscript entitled “GM-CSF protects macrophages from DNA damage by inducing differentiation”, Vico and colleagues investigate how growth factors in macrophages protect from DNA damage and eventually cell death. Overall, the study is rigorous, and the conclusions are mostly supported by the data. However, the main concern is all these studies are based on etoposide treatment of macrophages, a potent topoisomerase inhibitor that is far from physiological. The findings would benefit from showing convincing data with hydrogen peroxide, which may resemble more closely the reactive oxygen species that these cells are exposed to in a more physiological setting. Strengthening the data with H2O2 should be added when it comes to supporting the main conclusions (i.e. DNA damage response signalling by checking p-RPA and gammaH2AX, Bcl-2A1 induction, and differentiation). The other subject that should be discussed more about is whether these cells are actually resistant to DNA damage, or they have increased/rapid repair capabilities, and how differentiation plays a role in this process. What is the expression of the actual DNA repair machinery in proliferating vs differentiated cells?
There are numerous errors with figure calls that made review process confusing, some of which I am going to cite below, but the authors should check to the maximum detail to ensure correctness.
- Line 262-263: S1C and S1D do not show any gH2AX data, which may be omitted from the manuscript?
- Line 268: Fig 1D and 1E probably refer to Figure 1C and 1D
- Line 314: Fig S2A refers to S1C
- Line 352: Fig S2B refers to S1D
- Line 356: Fig 4B shows the data and not 4C
- Line 359: Fig 4D refers to 4C
- Line 361-362: Fig 4E refers to 4D
- Line 362: Fig S2C refers to S1E
- Line 391: Fig S2D refers to S1F
- Line 419: Fig S3 refers to S2
- Line 437: Fig S4A and S4B refer to S3A and S3B
- Line 439: Fig S5A refers to S4A
Methods would benefit from an additional Supplementary Materials Table including all the antibodies used, catalogue number, vendor, application for which antibody is used as well as dilution. This openness is crucial for the good progress in the scientific community.
Some other minor comments:
- Line 153: BMDM abbreviation hasn’t been defined earlier.
- Line 165: IP probably means PI?
- Line 275: H2A misses an X
- Line 315: etoposide should be replaced for “genotoxin” since results also apply to H2O2
Author Response

(The authors gave the same response as above.)

Round 2
Reviewer 1 Report
The authors have adequately addressed most of the comments and the paper has been significantly improved.
In response to the authors comment "Obviously, is not ethical to obtain human bone marrow tissue to perform experiments", I would like to inform the authors that human bone marrow is easily obtained during routine orthopaedic surgeries. Therefore, studies could be easily performed using human cells.
Reviewer 2 Report
The authors have partially addressed my issues, and the overall manuscript was quite ameliorated. In my opinion, it can be now published in its present form.
However, I suggested providing more representative pictures of SA-Bgal staining in Fig. 7B, cause the quantification does not correspond to what is seen on the provided pictures.
I also recommend changing the name of that staining for senescence-associated beta-galactosidase staining instead of beta-galactosidase staining as these two are not the same.
Reviewer 3 Report
The authors have addressed my concerns, and I am delighted to now recommend the manuscript for publication.
However, I advise to do some final spell checks, final figures calls and primarily check reference formatting (there is a mistake from reference 67 onwards, where reference list is added twice including the newly added references).
